# Hexafluorophosphate additive enables durable seawater oxidation at ampere-level current density

Xun He [1,2,3,6], Yongchao Yao[1,4,6], Limei Zhang[2,4], Hefeng Wang[3], Hong Tang[2], Wenlong Jiang[2], Yuchun Ren[2], Jue Nan[2], Yongsong Luo[1], Tongwei Wu [2] ✉, Fengming Luo[1] ✉, Bo Tang [3,5] ✉ & Xuping Sun [1,3] ✉

Direct seawater electrolysis at ampere-level current densities, powered by coastal/offshore renewables, is an attractive avenue for sustainable hydrogen production but is undermined by chloride-induced anode degradation. Here we demonstrate the use of hexafluorophosphate ($PF_6^-$) as an electrolyte additive to overcome this limitation, achieving prolonged operation for over 5,000 hours at $1\,A\,cm^{-2}$ and 2300 hours at $2\,A\,cm^{-2}$ using NiFe layered double hydroxide (LDH) as anode. Together with the experimental findings, $PF_6^-$ can intercalate into LDH interlayers and adsorb onto the electrode surface under an applied electric field, blocking $Cl^-$ and stabilizing Fe to prevent segregation. The constant-potential molecular dynamics simulations further reveal the accumulation of high surface concentrations of $PF_6^-$ on the electrode surface that can effectively exclude $Cl^-$, mitigating corrosion. Our work showcases synchronous interlayer and surface engineering by single non-oxygen anion species to enable $Cl^-$ rejection and marks a crucial step forward in seawater electrolysis.

Water electrolysis is pivotal to achieving global net-zero targets by 2050, yet the increasing hydrogen demand strains already scarce freshwater supplies[1-7]. As Earth's most abundant water source, seawater offers a promising alternative[8-11]. Alkaline seawater electrolysis bypasses energy-intensive desalination and extensive pretreatment, improving process efficiency[12-14]. When coupled with coastal or offshore renewable energy, it further reduces operational costs and facilitates scalable green hydrogen production[15,16]. Nonetheless, the high $Cl^-$ concentration (~0.5 M) promotes electrode corrosion through metal chloride–hydroxide pathways (Supplementary Note 1), accelerating anode degradation[17-20]. Moreover, while the oxygen evolution reaction (OER) is thermodynamically preferred in alkaline media, it is a four-electron process with sluggish kinetics. In contrast, the competing chlorine evolution reaction (ClER) proceeds via a two-electron pathway with lower kinetic barriers, generating toxic chlorine gas[21,22]. Although seawater alkalinization shifts the OER potential below that of ClER by 480 mV[23-26], achieving stable ampere-level current densities ($j$) remains a challenge due to increased resistance and persistent corrosion.

Despite incorporating precious metals, current anodes remain constrained by resource limitations, hindering the integration of catalytic activity, selectivity, and $Cl^-$ tolerance[27-31]. Recently, researches on transition-metal sulfides[32], phosphides[33], nitrides[34], or Ni(Co)Fe layered double hydroxides (LDHs)[35-40], have shown that in situ formation of oxyanion-rich layers (e.g., $SO_4^{2-}$, $PO_4^{3-}$, $NO_3^-$) or fabricating chloride-repellent coatings[37,38] can repel $Cl^-$, enabling efficient and

[1]Center for High Altitude Medicine, West China Hospital, Sichuan University, Chengdu, Sichuan, China. [2]Institute of Fundamental and Frontier Sciences, University of Electronic Science and Technology of China, Chengdu, Sichuan, China. [3]College of Chemistry, Chemical Engineering and Materials Science, Shandong Normal University, Jinan, Shandong, China. [4]Department of Laboratory Medicine/Clinical Laboratory Medicine Research Center, West China Hospital, Sichuan University, Chengdu, Sichuan, China. [5]Laoshan Laboratory, Qingdao, Shandong, China. [6]These authors contributed equally: Xun He, Yongchao Yao. ✉e-mail: twwu77@uestc.edu.cn; luofengming@wchscu.edu.cn; tangb@sdnu.edu.cn; xpsun@uestc.edu.cn

long-lasting seawater oxidation. LDHs are particularly attractive due to their layered structures, compositional tunability, and strong OER activity in alkaline media[41–44], but their resistance to $Cl^-$-induced degradation remains inadequate[36,37]. Strategies include creating physical shielding layers[35,45], adding oxyanion species ($SO_4^{2-}$, $PO_4^{3-}$) that can adsorb onto the anode surface[46,47], or forming surface chloride-immobilizing layers (AgCl, IrCl)[48,49] have been proposed to exclude $Cl^-$ via Coulombic repulsion or the common ion effect. The layered and positively charged nature of LDHs also supports anion intercalation, and numerous studies confirm that intercalated oxyanions can block chloride penetration[50–53]. Extending this concept, some studies have combined surface and interlayer engineering with dual species[54,55]. The Zou group[54] intercalated $CO_3^{2-}$ into CoFe LDH and attached graphene quantum dots on the surface, achieving 2800 h of long-term durability at ~1.25 A cm$^{-2}$ in simulated seawater. Our group[55] further introduced $OH^-$-enriching $Cr_2O_3$ onto NiFe LDH surfaces and $Cl^-$-repelling $CrO_4^{2-}$ into the interlayers, enabling 1000 h at 1 A cm$^{-2}$ followed by 1500 h at 2 A cm$^{-2}$. Despite these advancements indicating considerable progress, developing simple and effective strategies to sustain prolonged electrolysis (>5000 h) under high $j$ for anode materials is still a major challenge.

In this work, a NiFe LDH anode, operated with hexafluorophosphate ($PF_6^-$) as an electrolyte additive, delivers stable operation for over 5000 h at 1 A cm$^{-2}$ and 2300 h at 2 A cm$^{-2}$. Ex situ/in situ evidences show that $PF_6^-$, driven by the electric field, can facilely intercalate into the LDH layers and adsorb onto the surface, facilitating a single-ion-driven surface-interlayer synergy that excludes $Cl^-$. This additive also improves the durability of CoFe LDH, enabling continuous operation at 2 A cm$^{-2}$ for 1200 h, thereby rendering CoFe LDH/NiFe LDH as top contenders for industrial-level seawater electrolysis. Constant-potential molecular dynamics simulations further verify that surface-accumulated $PF_6^-$ efficiently excludes $Cl^-$. Beyond its durability, the NiFe LDH anode coupled with a Pt/C cathode also delivers notable performance in an alkaline seawater electrolyzer.

## Results

### Enhanced alkaline seawater oxidation (ASO) performance with $PF_6^-$ additive

The NiFe LDH nanosheet array, consisting of two-dimensional (2D) layers, was fabricated on nickel foam (NiFe LDH/NF) through a one-step hydrothermal approach[56]. Extensive characterizations via X-ray diffraction (XRD), scanning electron microscopy (SEM), transmission electron microscopy (TEM), high-resolution TEM (HRTEM), and high-angle annular dark field scanning TEM (HAADF-STEM) verify the successful synthesis of NiFe LDH/NF with a Ni:Fe atomic ratio of ~3:1 (Supplementary Figs. 1–5, Supplementary Table 1).

A standard three-electrode configuration was employed to evaluate the OER performance of NiFe LDH/NF in alkaline seawater, either containing 20 mM $PF_6^-$ or without any additive. As shown in the polarization curves (Fig. 1a, Supplementary Fig. 6), the presence of $PF_6^-$ reduces the overpotentials required to obtain the $j$ of 200, 500, and 1000 mA cm$^{-2}$ (209, 247, 282 mV) compared to the corresponding values of 282, 315, and 343 mV in $PF_6^-$-free seawater. NiFe LDH/NF in $PF_6^-$-containing seawater also outperforms bare NF, benchmark $RuO_2$ and $IrO_2$ catalysts, ranking among the top-tier catalysts reported to date (Supplementary Table 2). Turnover frequency (TOF) analysis (inset of Fig. 1a, Supplementary Fig. 7) further shows enhanced intrinsic activity in the presence of $PF_6^-$. Electrochemical double-layer capacitance ($C_{dl}$) measurements (Supplementary Fig. 8) indicate a higher electrochemically active surface area (ECSA) with $PF_6^-$, in line with its enhanced OER activity. Even after normalizing $j$ to ECSA (Supplementary Fig. 9), NiFe LDH/NF in $PF_6^-$-containing seawater maintains higher activity than in the $PF_6^-$-free counterpart. Tafel slope analysis based on polarization curves obtained at a slow scan rate of 1 mV s$^{-1}$ (Supplementary Fig. 10, Fig. 1b) confirms that $PF_6^-$ addition

improves the reaction kinetics for NiFe LDH/NF, outperforming $RuO_2$ and $IrO_2$. Electrochemical impedance spectroscopy over the potential range of 1.405–1.485 V (Supplementary Fig. 11) further reveals that NiFe LDH/NF in $PF_6^-$-containing seawater exhibits lower charge-transfer resistance than in $PF_6^-$-free seawater, supporting accelerated interfacial electron transfer. To assess corrosion resistance, Tafel scanning was performed (Fig. 1c), showing that the corrosion potential of NiFe LDH/NF shifts positively in $PF_6^-$-containing seawater, indicative of improved corrosion stability. Moreover, Faradaic efficiency for oxygen evolution at 1 A cm$^{-2}$ reaches nearly 100% with NiFe LDH/NF in $PF_6^-$-containing seawater (Supplementary Fig. 12), confirming selective four-electron $O_2$ evolution. At an industrial-level $j$ of 1.0 A cm$^{-2}$, NiFe LDH/NF in $PF_6^-$-containing seawater operates stably for over 5000 h (Fig. 1d), a 41.6-fold improvement in stark contrast to its rapid failure within 120 hours in $PF_6^-$-free seawater (inset of Fig. 1d). Ultraviolet–visible spectroscopy (Supplementary Fig. 13) verifies minimal active chlorine generation with $PF_6^-$ present, in stark contrast to high chlorine levels in the $PF_6^-$-free seawater electrolyte. Even under a higher $j$ of 2.0 A cm$^{-2}$, NiFe LDH/NF demonstrates stable operation in $PF_6^-$-containing seawater for over 2300 h (Fig. 1e). In light of these results, using NiFe LDH/NF anode with $PF_6^-$ as an electrolyte additive facilitates a notable advance for seawater electrolysis at high $j$, beyond that of currently explored anionic species (Supplementary Table 3), and showing promise for industrial applications.

### Investigation of activity enhancement and corrosion resistance

Operando Raman spectroscopy was performed using a custom-designed electrochemical cell (Supplementary Fig. 14a) to monitor structural changes for NiFe LDH from open-circuit potential (OCP) to 1.75 V $vs.$ the reversible hydrogen electrode (RHE). In $PF_6^-$-free seawater, Raman spectra collected at potentials below 1.35 V show peaks at 454 and 527 cm$^{-1}$, assigned to the $A_{1g}$ stretching modes of Ni–OH and Ni–O, respectively (Fig. 2a). Upon increasing the applied potential, two additional peaks emerge at 475 and 556 cm$^{-1}$, corresponding to the $E_g$ bending and $A_{1g}$ stretching vibrations of Ni–O for NiOOH[44,57–59]. Concurrently, the disappearance of the Ni–OH lattice vibration (~300 cm$^{-1}$) and attenuation of signals associated with $CO_3^{2-}/NO_3^-$ species (~727 cm$^{-1}$) further support the occurrence of dehydration and oxidation processes during OER[60–62]. In $PF_6^-$-containing seawater, apart from the progressive blue shifts of Ni–O bands (454/527 to 475/556 cm$^{-1}$), an additional $E_g$ vibration of $PF_6^-$ is observed at increasing potentials, potentially due to its intercalation into LDH layers. As the applied potential continues to rise, the intensification of the $A_{1g}$ vibration indicates extensive $PF_6^-$ accumulation on the electrode surface driven by a stronger applied electric field (Fig. 2b, Supplementary Fig. 14b). To verify electric-field-driven intercalation, NiFe LDH/NF electrode was held at 1.15 V for 10 min in $PF_6^-$-containing seawater. Structural analyses, including XRD, SEM, TEM, HRTEM, and HAADF-STEM (Supplementary Figs. 15–19), collectively confirm the successful incorporation of $PF_6^-$ into the NiFe LDH interlayers. Given its large ionic radius, $PF_6^-$ displaces native interlayer anions, expanding the interlayer spacing, exposing more active sites, thus promoting OER activity. Raman spectroscopy after intercalation (Supplementary Fig. 20) further confirms this process, as evidenced by the disappearance of characteristic $CO_3^{2-}/NO_3^-$ peaks and the emergence of a pronounced $E_g$ vibration mode of $PF_6^-$. Additionally, we observed a slight blue shift for NiFe LDH Raman peaks following $PF_6^-$ intercalation (Supplementary Fig. 21), contrary to the usual red shifts induced by cation-driven Ni–O bond elongation[58,63,64]. Operando Raman spectroscopy of $PF_6^-$-intercalated NiFe LDH in $PF_6^-$-free seawater (Fig. 2c) shows delayed γ-NiOOH formation, consistent with the observations in $PF_6^-$-containing seawater. With increasing potential, partial deintercalation of $PF_6^-$ occurs, accompanied by interlayer contraction due to shortening of Ni–O bonds. A subsequent chronopotentiometry test at 2 A cm$^{-2}$ shows that $PF_6^-$-intercalated NiFe LDH/NF maintains stable operation for about

520 h in $PF_6^-$-free seawater (Supplementary Fig. 22), substantially less than the 2300 h achieved in $PF_6^-$-containing seawater. These observations underscore the crucial role of surface-accumulated $PF_6^-$ in enhancing resistance to chloride-induced corrosion under prolonged high $j$ conditions.

X-ray photoelectron spectroscopy analyses also confirm the successful incorporation of $PF_6^-$, quantified at 5.93 wt.% by inductively coupled plasma mass spectrometry (ICP-MS; Supplementary Table 4), with distinct P 2$p$ and F 1$s$ peaks (Supplementary Fig. 23a, b), alongside shifts toward lower and higher binding energies for Ni 2$p$ and Fe 2$p$, respectively (Supplementary Fig. 23c, d). X-ray absorption near-edge structure (XANES) analyses further reinforce these findings, revealing a reduced oxidation state for Ni and elevated oxidation state for Fe after $PF_6^-$ intercalation (Fig. 2d, e). First-derivative Fe K-edge spectra of the $PF_6^-$-intercalated NiFe LDH reveal enhanced Fe–M (M = Fe or Ni) coordination (Fig. 2f), supporting the idea that Fe experiences stronger

bonding and electron withdrawal. Conversely, Ni–O coordination weakens (Supplementary Fig. 24), and Ni becomes more reduced, suggesting that $PF_6^-$ does not directly remove electrons from Ni but instead facilitates electron transfer from Fe to Ni, which potentially account for the delayed γ-NiOOH formation observed for $PF_6^-$-intercalated NiFe LDH and NiFe LDH in $PF_6^-$-containing seawater. Furthermore, the slight contraction of the Ni–O bond upon $PF_6^-$ intercalation accounts for the blue shift observed in Raman spectra (Supplementary Fig. 21).

We quantified Fe segregation after 120-hour chronoamperometry test at 1 A cm$^{-2}$. The Fe concentration leached from NiFe LDH/NF in $PF_6^-$-containing seawater is notably lower (0.291 μg mL$^{-1}$) compared to $PF_6^-$-free seawater (1.5 μg mL$^{-1}$). Correspondingly, the Fe retained in $PF_6^-$-containing conditions is markedly higher (263.5 μg cm$^{-2}$ vs. 14.5 μg cm$^{-2}$; Supplementary Fig. 25). Time-of-flight secondary ion mass spectrometry (TOF-SIMS) further confirms the retention of Fe,

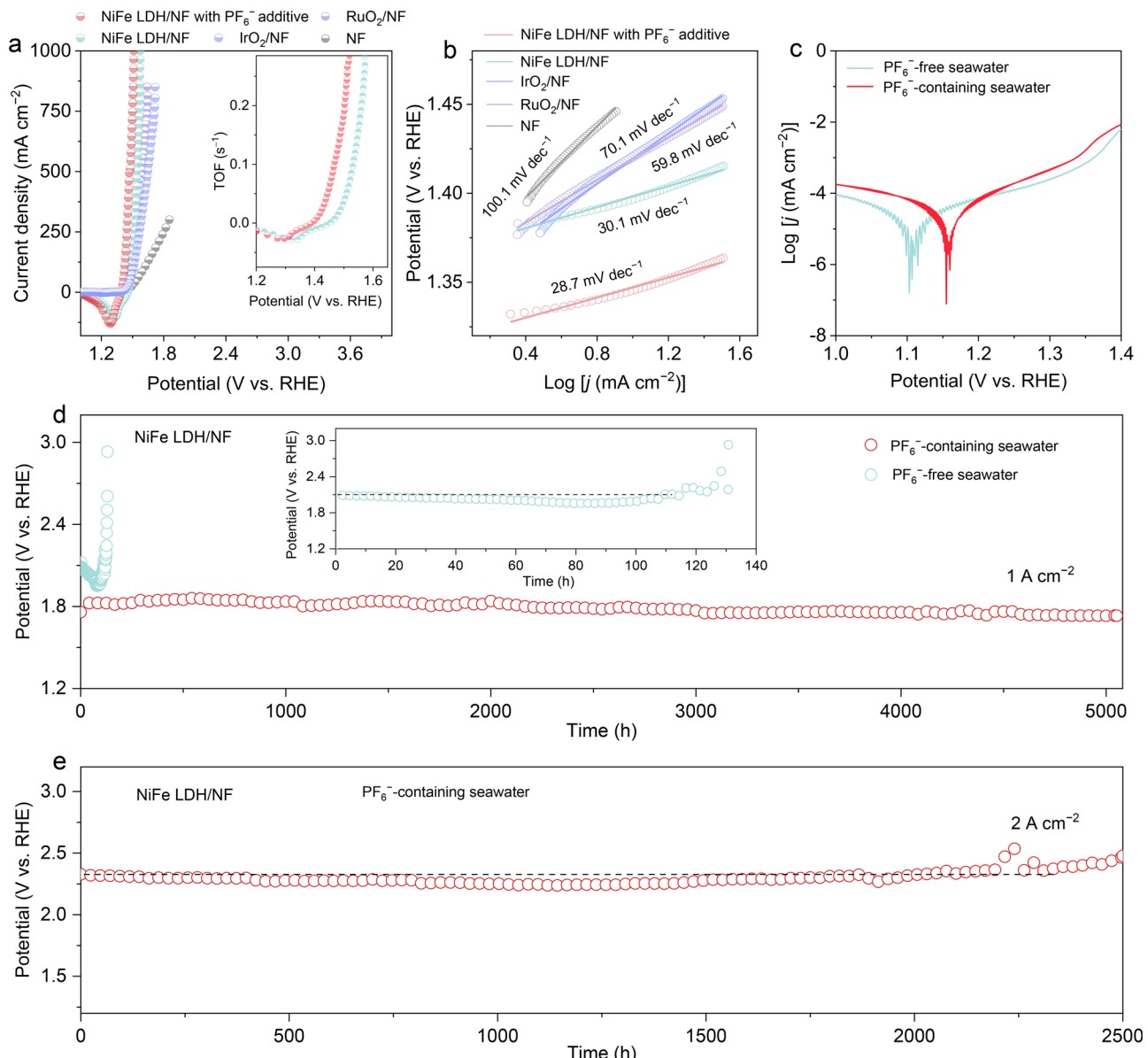

**Fig. 1 | ASO performance. a** Polarization curves at a scan rate of 10 mV s$^{-1}$ and **b** corresponding Tafel slopes of NiFe LDH/NF in $PF_6^-$-free and $PF_6^-$-containing seawater, alongside RuO$_2$/NF, IrO$_2$/NF and NF in $PF_6^-$-free seawater with 100% $iR$ correction. **c** Corrosion behavior curves for NiFe LDH/NF in $PF_6^-$-free and $PF_6^-$-containing seawater. **d** Chronopotentiometry curves at a $j$ of 1 A cm$^{-2}$ for NiFe LDH/

NF in $PF_6^-$-free and $PF_6^-$-containing seawater without $iR$ correction. The inset magnifies a section of the chronopotentiometry curve for NiFe LDH/NF in $PF_6^-$-free seawater. **e** Chronopotentiometry curve at a $j$ of 2 A cm$^{-2}$ for NiFe LDH/NF in $PF_6^-$-containing seawater without $iR$ correction. Source data are provided as a Source Data file.

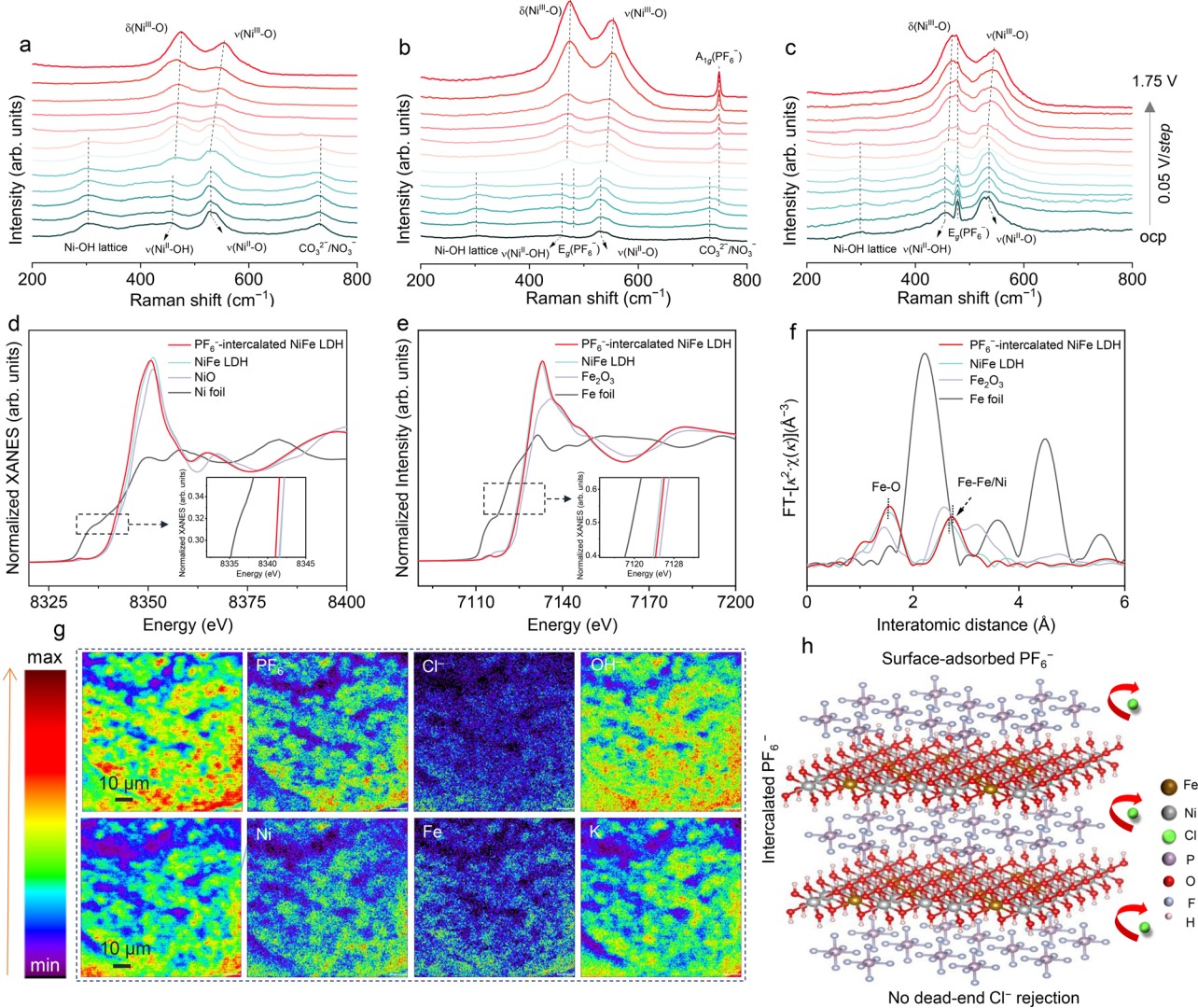

**Fig. 2 | Exploring the role of PF$_6^-$.** Operando Raman spectra collected from OCP to 1.75 V *vs.* RHE for NiFe LDH/NF in **a** PF$_6^-$-free and **b** PF$_6^-$-containing seawater. **c** Operando Raman spectra under the same potential range for PF$_6^-$-intercalated NiFe LDH/NF in PF$_6^-$-free seawater. **d** Normalized Ni K-edge XANES spectra for PF$_6^-$-intercalated NiFe LDH/NF, NiFe LDH, NiO, and Ni foil. **e** Normalized Fe K-edge XANES spectra for PF$_6^-$-intercalated NiFe LDH, NiFe LDH, Fe$_2$O$_3$, and Fe foil. **f** First-derivative Fe K-edge spectra of PF$_6^-$-intercalated NiFe LDH, NiFe LDH, Fe$_2$O$_3$, and Fe foil. **g** TOF-SIMS mapping of PF$_6^-$, Cl$^-$, OH$^-$, Ni, Fe, and K fragments on PF$_6^-$-intercalated NiFe LDH/NF following a 120-h test. **h** Schematic of the proposed PF$_6^-$ intercalation and surface adsorption mechanisms for Cl$^-$ rejection. Source data are provided as a Source Data file.

abundant PF$_6^-$, and minimal Cl$^-$, verifying the effective Cl$^-$ exclusion by PF$_6^-$ (Fig. 2g). Morphological characterizations reveal that NiFe LDH/NF in PF$_6^-$-containing seawater preserves its NF skeleton and 2D nanosheet arrays, while the PF$_6^-$-free electrode suffers severe structural failure after 120 hours (Supplementary Figs. 26, 27), affirming the efficacy of PF$_6^-$ in enhancing mechanical durability and mitigating chloride-induced corrosion. Furthermore, EDS mapping and XPS analyses (Supplementary Figs. 28, 29) conducted after the long-term test confirm the presence of P, F, Ni, Fe, and O throughout the nanosheets, alongside extensive NiOOH formation at the surface, also supporting that PF$_6^-$ incorporation stabilizes Fe and suppresses its leaching. Raman spectroscopy further verifies the surface phase transformation and the retention of PF$_6^-$ within the interlayers (Supplementary Fig. 30).

Through combined ex situ and in situ analyses, we verify that PF$_6^-$ can intercalate into the NiFe LDH interlayers and accumulate on the electrode surface under applied electric field (Fig. 2h). This dual regulation of interlayer species and surface adsorption by a single non-oxygen anion contributes to Fe stabilization by suppressing

segregation, and efficiently excludes Cl$^-$, enabling durable ampere-level seawater oxidation.

## Catalytic durability with PF$_6^-$ additive: insights from experiment and molecular dynamics

Our strategy is also applicable to other LDHs for ASO catalysis, as demonstrated using CoFe LDH as a model. Operando Raman spectra collected from OCP to 1.70 V also validate the intercalation and surface adsorption behavior of PF$_6^-$ (Supplementary Fig. 31), with additional evidence from XPS P 2*p* and F 1*s* spectra supporting PF$_6^-$ incorporation (Supplementary Fig. 32). Electrochemical tests show that CoFe LDH/NF in PF$_6^-$-containing seawater exhibits lower overpotentials than in PF$_6^-$-free seawater (Supplementary Fig. 33), achieving 1 A cm$^{-2}$ at only 373 mV, outperforming many reported CoFe-based ASO anodes (Supplementary Table 5). Stability tests at 2.0 A cm$^{-2}$ show over 1200 h of stable operation in the presence of PF$_6^-$, compared to failure after 65 h without it (Fig. 3a), representing an 18.5-fold improvement. The long-term durability achieved under high current density also enders CoFe LDH/NF as the most robust CoFe-based ASO anode currently

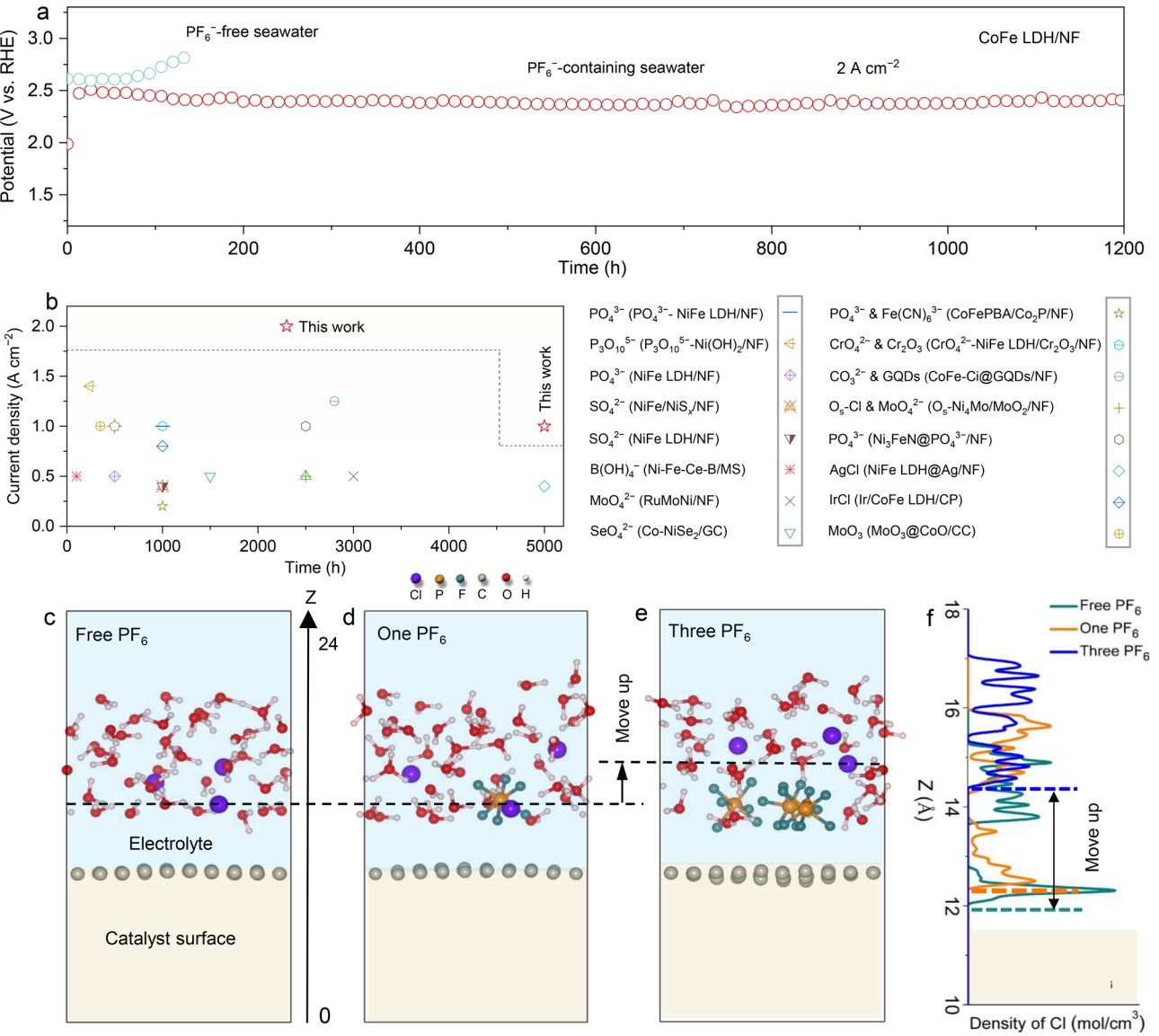

**Fig. 3 | Electrochemical and theoretical insights into PF$_6^-$ enhancing ASO stability. a** Chronopotentiometry curves at a $j$ of 2 A cm$^{-2}$ for CoFe LDH/NF in PF$_6^-$-free and PF$_6^-$-containing seawater without $iR$ correction. **b** Comparison of various state-of-the-art electrodes and Cl$^-$ repelling strategies for ASO. **c**–**e** Theoretical calculations for the interactions between PF$_6^-$ and Cl$^-$ at the electrolyte/electrode interface at a positive potential of 1.5 V. **f** The density distribution of Cl$^-$ at the electrolyte/electrode interface. Source data are provided as a Source Data file.

known (Supplementary Table 6). Notably, the PF$_6^-$ additive enables synchronous interlayer and surface engineering of benchmark CoFe LDH/NiFe LDH catalysts under an applied electric field, offering a simple and more effective Cl$^-$ exclusion strategy compared to physical shielding, surface chloride immobilization, oxyanion-based electro-static repulsion, or dual-species interlayer/surface modifications (Supplementary Table 7, Fig. 3b).

To further elucidate the mechanism by which PF$_6^-$ enhances corrosion resistance, we employed advanced constant-potential molecular dynamics (CPMD) simulations to investigate the interactions between PF$_6^-$ and Cl$^-$ at the electrolyte-electrode interface under a positive potential of U = 1.5 V[65]. This approach provides an accurate simulation of atomic-scale dynamics at the electrocatalyst/electrolyte interface under operational conditions[65]. The calculation results indicate that the introduction of a single PF$_6^-$ into the electrolyte has no marked effect on Cl$^-$ at the electrode surface (Fig. 3c, d), because the density distribution of Cl$^-$ on the electrode surface does not change considerably. Further increasing the number of PF$_6^-$ to three generates a higher concentration of PF$_6^-$ on the electrode surface, and we

observe that the density of Cl$^-$ moves substantially away from the electrode surface by about 2 Å (Fig. 3e, f). These results support the operando Raman spectroscopy and electrochemical durability results, where the accumulation of PF$_6^-$ at the catalyst/electrolyte interface under high $j$ correlates with enhanced long-term stability.

## Practical electrolysis applications

To evaluate the practical application, we constructed a membrane electrode assembly (MEA) electrolyzer with an anion exchange membrane (AEM, PiperION-A60) (Fig. 4a). In this configuration, anions and water migrate through the AEM, facilitating electron transfer at the NiFe LDH/NF anode and the Pt/C/NF cathode. Concurrently, H$_2$, O$_2$, and discharged seawater, are expelled from the electrolyzer chamber. For comparison, a RuO$_2$/NF||Pt/C/NF configuration was also tested. The NiFe LDH/NF||Pt/C/NF electrolyzer employs a dual-feed approach with 6 M KOH+ seawater served as the catholyte and the anolyte consisted of 6 M KOH+ seawater with the addition of PF$_6^-$. In contrast, the RuO$_2$/NF||Pt/C/NF electrolyzer uses 6 M KOH+ seawater as both the anolyte and catholyte. All experiments were meticulously conducted

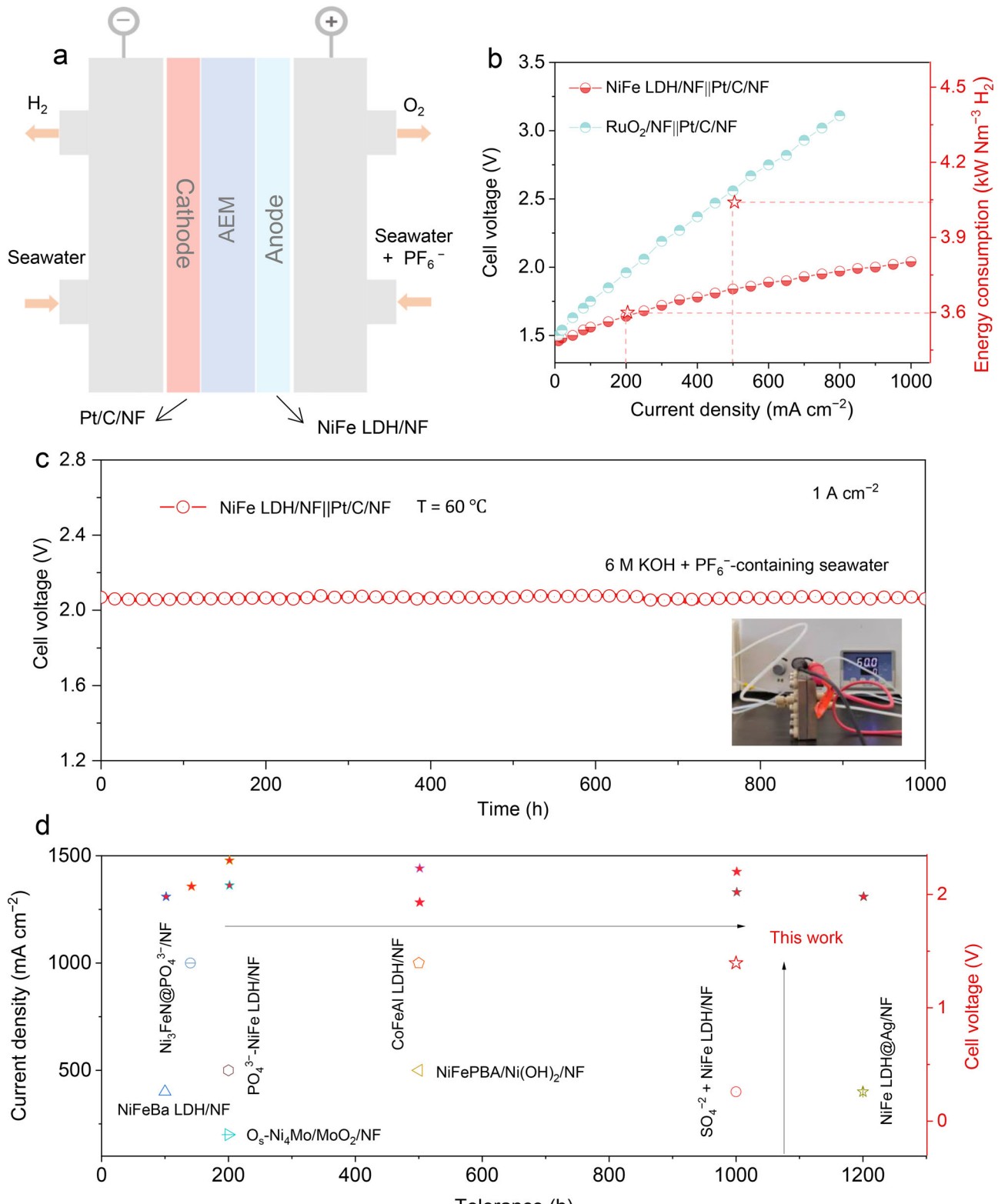

**Fig. 4 | Flow-cell assessment of NiFe LDH/NF with PF₆⁻ additives. a** Schematic of the flow electrolytic cell with asymmetric seawater feed. **b** Polarization curves of NiFe LDH/NF||Pt/C/NF and RuO₂/NF||Pt/C/NF without *iR* correction, with energy consumption values indicated for NiFe LDH/NF||Pt/C/NF. **c** Chronopotentiometry curve at a *j* of 1.0 A cm⁻² for NiFe LDH/NF||Pt/C/NF. Inset shows a photograph of two electrode electrolyzer. **d** Comparison of cell voltages and long-term electrolysis tolerance for NiFe LDH/NF with PF₆⁻ additives against recently reported ASO catalysts. Source data are provided as a Source Data file.

at a high temperature of 60 °C to simulate industrial operational conditions. The performance metrics of the MEA with NiFe LDH/NF∥Pt/C/NF exhibits notable electrocatalytic activity, outperforming that of the RuO$_2$/NF∥Pt/C/NF counterpart (Fig. 4b). Specifically, the NiFe LDH/NF∥Pt/C/NF electrolyzer requires only 2.02 V to achieve a $j$ of 1.0 A cm$^{-2}$. Furthermore, the energy consumption at the $j$ of 0.2 and 0.5 A cm$^{-2}$ is calculated to be 3.60 and 4.04 kWh m$^{-3}$ H$_2$, respectively, outperforming conventional electrolyzer that demand 4.47 kWh m$^{-3}$ at 0.2 A cm$^{-2}$. Stability tests were carried out to evaluate the long-term robustness of the electrocatalytic system. The NiFe LDH/NF∥Pt/C/NF electrolyzer operating in 6 M KOH + seawater with PF$_6^-$ additives maintain stable performance for over 1000 h at a $j$ of 1.0 A cm$^{-2}$ (Fig. 4c). In contrast, the same MEA electrolyzer in PF$_6^-$-free seawater experiences rapid degradation after only 31 hours (Supplementary Fig. 34), further underscoring the critical role of PF$_6^-$ in sustaining electrode stability. A preliminary economic analysis (Supplementary Note 2) indicates that, under a typical electricity rate of US\$0.02 kWh$^{-1}$, the hydrogen production cost is approximately US\$1.07 kg$^{-1}$ H$_2$ at 1.0 A cm$^{-2}$ and 2.02 V. Additionally, Supplementary Table 8 and Fig. 4d illustrates that the incorporation of PF$_6^-$ additives into the NiFe LDH/NF anode not only provides a competitive cell voltage at high $j$ but also ensures prolonged electrolysis stability. Specifically, the MEA electrolyzer with NiFe LDH/NF anode and PF$_6^-$ additives exhibit outstanding activity and durability, paving an important step toward practical deployment of seawater electrolysis for hydrogen production.

## Discussion

Our study identifies PF$_6^-$ as a crucial electrolyte additive that markedly enhances the durability of NiFe LDH anode, achieving over 5000 h of stable operation at 1 A cm$^{-2}$ and 2300 h at 2 A cm$^{-2}$. Comprehensive ex situ/in situ studies and CPMD simulations reveal the mechanisms behind this improvement. Firstly, under a low applied electric field, PF$_6^-$ intercalates into the LDH layers, expanding interlayer spacing and exposing more active sites, thereby boosting ASO activity. Concurrently, PF$_6^-$ weakly coordinates with Fe sites, preventing segregation and preserving electrode integrity. Secondly, PF$_6^-$ also adsorbs and accumulates on the electrode surfaces under a high applied electric field, effectively blocking Cl$^-$. In a two-electrode MEA electrolyzer, this configuration requires ~19.46% less electrical energy than conventional electrolyzer and maintains electrolysis durability for over 1000 h. This study pioneers the efficacy of single-anion engineering in enabling interlayer/surface modifications, providing LDH structures with structural integrity and Cl$^-$-repelling capabilities, thereby paving the way for sustainable and large-scale hydrogen production.

## Methods

### Materials

All chemicals used in this study are listed below with their respective purity and sources. Nickel nitrate hexahydrate (Ni(NO$_3$)$_2$·6H$_2$O, AR), cobalt nitrate hexahydrate (Co(NO$_3$)$_2$·6H$_2$O, AR), potassium hydroxide (KOH, 96 wt.%), ammonium fluoride (NH$_4$F, AR), urea (CO(NH$_2$)$_2$, AR), potassium hexafluorophosphate (KPF$_6$, AR), ruthenium oxide (RuO$_2$, AR), iridium oxide (IrO$_2$, AR), platinum on carbon (Pt/C, 20 wt.%), Nafion (5 wt.%), sodium hypochlorite (NaClO, AR), and iron nitrate nonahydrate (Fe(NO$_3$)$_3$·9H$_2$O, AR) were obtained from Aladdin Industrial Co. Hydrochloric acid (HCl, 98 wt.%) and anhydrous ethanol were sourced from Beijing Chemical Corp. Nickel foam (NF, 0.2 mm thick) was purchased from Qingyuan Metal Materials Co., Ltd. in Xingtai. Seawater samples were collected from Huangdao District, Qingdao City. All aqueous solutions were prepared using deionized water with a resistivity of 18.3 MΩ·cm.

### Synthesis of NiFe LDH/NF

NiFe LDH was grown on NF (2 cm × 3 cm) via a hydrothermal process. An aqueous solution containing 0.7 mmol Fe(NO$_3$)$_3$·9H$_2$O, 2.1 mmol Ni(NO$_3$)$_2$·6H$_2$O, 4 mmol NH$_4$F, and 10 mmol urea was transferred to a Teflon-lined autoclave along with NF and heated at 120 °C for 6 h. After cooling to room temperature, the NiFe LDH/NF was rinsed with deionized water, dried at 60 °C. The loading mass of NiFe LDH is ~1.93 mg cm$^{-2}$.

### Synthesis of PF$_6^-$-intercalated NiFe LDH/NF

NiFe LDH/NF electrode was held at 1.15 V for 10 min in PF$_6^-$-containing seawater to obtain PF$_6^-$-intercalated NiFe LDH/NF, with Hg/HgO and graphite serving as reference and counter electrodes, respectively. The sample was then removed, rinsed with deionized water, dried at 60 °C. The resulting material has a loading mass of ~2.02 mg cm$^{-2}$.

### Synthesis of CoFe LDH/NF

CoFe LDH was prepared on NF (2 cm × 3 cm) by dissolving 3 mmol Co(NO$_3$)$_2$·6H$_2$O, 1 mmol Fe(NO$_3$)$_3$·9H$_2$O, and 3 mmol urea in 30 mL deionized water. This mixture and the nickel foam substrate were placed in a Teflon-lined autoclave and heated at 120 °C for 6 h. After cooling, the CoFe LDH/NF was thoroughly rinsed, dried at 60 °C. The loading mass of CoFe LDH is ~1.59 mg cm$^{-2}$.

### Preparation of IrO$_2$/NF and RuO$_2$/NF

The IrO$_2$/NF and RuO$_2$/NF electrodes were prepared following the procedures reported in our previous work without modification[21]. Commercial RuO$_2$ and IrO$_2$ powders were loaded onto NF with the same mass loading as that of NiFe LDH/NF.

### Characterizations

The crystal phases of the synthesized materials were identified using X-ray diffraction (XRD) with a Bruker D8 Advance instrument. Surface and structural morphologies were examined via scanning electron microscopy (SEM) on a Gemini SEM 300 (ZEISS) and transmission electron microscopy (TEM) alongside high-resolution TEM (HRTEM) using a JEM-F200 (JEOL). Elemental and chemical state analyses were carried out using X-ray photoelectron spectroscopy (XPS) on an ESCALABMK II spectrometer equipped with an Mg source. X-ray absorption fine structure (XAFS) data were collected at room temperature at the Shanghai Synchrotron Radiation Facility (beamline BL11B). The metal content of the catalysts and the extent of Fe leaching in alkaline seawater were measured using inductively coupled plasma mass spectrometry (ICP-MS, Agilent 5110). Optical absorbance was assessed with a SHIMADZU UV-1800 spectrophotometer, and detailed surface composition was analyzed using time-of-flight secondary ion mass spectrometry (TOF-SIMS) on a PHI TRIFT V nano TOF system.

### Electrochemical tests

ASO evaluations were performed at room temperature using a CHI 660E electrochemical workstation, and electrode stability was tested with a LANHE battery tester (Wuhan, China) under steady current conditions. A typical three-electrode arrangement was implemented, using NiFe LDH/NF, CoFe LDH/NF, RuO$_2$/NF, IrO$_2$/NF, or NF as the working electrodes. The reference electrode was Hg/HgO, and a graphite rod served as the counter electrode. The electrolytes used were either 1 M KOH + seawater or seawater containing 20 mM PF$_6^-$, with a total volume of 50 mL. To reduce magnesium and calcium ion concentrations, natural seawater was treated with sodium carbonate (Na$_2$CO$_3$, AR) prior to use[21]. To prepare 1 M KOH + 20 mM PF$_6^-$ seawater electrolytes, 56.11 g of KOH and 3.69 g of KPF$_6$ were dissolved in 1 L of treated seawater, stirred, and sonicated. The electrolytes were used within 1 h of preparation, kept at room temperature. The $pH$ was measured to be 13.97 ± 0.11. Electrode potentials were recalibrated against the RHE using the equation: $E_{RHE} = E_{Hg/HgO} + 0.098 + 0.059 × pH$. The Hg/HgO electrode was calibrated in H$_2$ using a Pt wire as the working electrode. The iR-compensated potential ($E_{corr}$) was determined using the relation $E_{corr} = E - iR$, where E is the

measured potential, $i$ is the applied current, and $R$ is the solution resistance determined at OCP, measured to be $1.544 \pm 0.01\,\Omega$. Electrochemical impedance spectroscopy measurements were performed over a frequency range of 10 kHz to 0.01 Hz with a perturbation amplitude of 5 mV. Prior to long-term durability evaluation, the electrode was subjected to chronopotentiometric activation to facilitate $PF_6^-$ intercalation. Throughout the test, alkaline seawater was continuously replenished to counteract electrolyte loss.

### TOF calculation

TOF was determined using the formula TOF = $Aj/(4Fm)$, where A represents the geometric of the electrode, $j$ is the current density, 4 corresponds to the number of electrons per mole of $O_2$ produced, and F is the Faradaic constant (96,485 C mol$^{-1}$). The parameter m denotes the active site concentration in moles. To calculate m, the slope of the oxidation peak current versus scan rate was analyzed using the equation Slope = $n^2F^2m/4RT$, with $n = 1$ assumed for a single-electron process.

### ECSA and ECSA-normalized $j$

The specific activity was determined by normalizing the $j$ to ECSA. The ECSA was calculated based on the $C_{dl}$, measured through CV curves from 1.09 to 1.19 V vs. RHE without $iR$ correction, where no Faradaic current occurs. $C_{dl}$ was plotted as $\Delta j/2$ at 1.14 V vs. RHE against scan rates to obtain the slope. ECSA was then derived using the formula ECSA = $AC_{dl}/C_s$, where $C_s$ is the specific capacitance (0.04 mF·cm$^{-2}$)[55].

### Operando Raman tests

Raman spectra were recorded using a LabRAM HR Evolution confocal Raman spectrometer with a 532 nm laser and a 50× objective lens. Operando experiments were conducted using a custom-designed Raman cell, where NiFe LDH, $PF_6^-$-intercalated NiFe LDH or CoFe LDH served as the working electrode. A platinum wire was utilized as the counter electrode, and Hg/HgO acted as the reference electrode. The electrolyte solution comprised 1 M KOH + seawater or $PF_6^-$-containing seawater. Voltage-time tests were executed on a CHI 660E electrochemical workstation at OCP−1.75 V vs. RHE (step: 0.05 V).

### Constant potential molecular dynamics calculations

Constant potential molecular dynamics (CPMD) is an effective methodology for examining the dynamics of electrochemical interfaces in explicit solvent environments under fixed potential conditions. This approach is well-suited for systems of moderate size, allowing comprehensive investigation of interfacial properties and electrochemical interactions[65]. The simulation procedure began by arranging 40 explicit water molecules to create a density of 1 g cm$^{-3}$. Following this, three Cl$^-$ were placed randomly in the aqueous phase to ensure uniform distribution. Finally, we achieved the variation of different concentrations by introducing different numbers of $PF_6^-$ on the surface of the catalyst. In our CPMD simulations, Langevin dynamics with a friction coefficient of 0.2 was utilized to regulate the system temperature at 300 K[66]. A simulation timestep of 1 femtosecond (fs) was adopted, with hydrogen atoms assigned a mass of 2 atomic mass units (u). The simulation trajectories were conducted with eigenstate settings of $1.0e^{-4}$, density settings of $1.0e^{-5}$, and energy settings of $1e^{-6}$, ensuring accurate representation of the electrochemical processes.

### Fabrication of MEA

The MEA was constructed using NiFe LDH/NF as the anode and Pt/C/NF with a geometric area of 1 cm$^2$ as the cathode. An anion exchange membrane (60 μm thickness, 1.2 cm × 1.2 cm in size) was pre-treated in 1 M KOH for over 12 h to convert it into the hydroxide form, followed by thorough rinsing with deionized water. During operation, alkaline seawater electrolyte was continuously circulated through the cell at a constant flow rate of 50 mL min$^{-1}$, and full-cell performance was evaluated at 60 °C using a GW Instek PSW 80-13.5.

## Data availability

The source data generated in this study are provided in the Source Data file. Source data are provided with this paper.

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

## Acknowledgements

X.S. thanks the funding support from the Free Exploration Project of Frontier Technology for Laoshan Laboratory (No. 16-02). F.L. acknowledges the funding support from the Science and Technology Program of Tibet (No. XZ202201ZY0002G). T.W. acknowledges the funding support from the Natural Science Foundation of China (No. 52202214). The numerical calculations in this paper have been done on Computing Center in Xi'an. The authors thank BL11B beamline of the Shanghai Synchrotron Radiation Facility (SSRF) for providing the XAFS beamtime.

## Author contributions

X.S. designed this research. X.H. and X.S. wrote the manuscript. X.H. and Y.Y. conducted material synthesis, characterizations, and performance tests. X.H., Y.Y. and L.Z. conceived and completed all the schematic drawings. H.W. performed the operando Raman tests. Y.L. performed the SEM tests. H.T., W.J., Y.R. and J.N. participated in discussions. T.W. conducted theoretical calculations. F.L., B.T. and X.S. supervised the research. All authors contributed and reviewed the manuscript.

## Competing interests

The authors declare no competing interests.
