## [Transparent Peer Review file · Nature Communications]

Hexafluorophosphate additive enables durable seawater oxidation at ampere-level current density

Corresponding Author: Professor Xuping Sun

Version 0:

Reviewer comments:

Reviewer #1

(Remarks to the Author)

In this study, Sun and coworker report the role of hexafluorophosphate as an effective additive to improve direct seawater hydrolysis. They demonstrate that the addition of hexafluorophosphate in electrolytes can intercalate into interlayers of nonprecious transition metal LDH electrocatalysts, adsorb onto electrode surfaces, and accumulate to repel chloride ions. They also combine molecular dynamics simulations and experimental analysis to support the claims. In my opinion, the introduction of anion additive in electrolyte presents a novel, facile approach to well address the stability problem in seawater oxidation, presenting a solid progress in this field that is increasing research interest currently. Thus, I'd like to recommend publication of this work in Nature Communications after minor revision by considering the comments and concerns below.

- 1) Authors state that PF_6^- can intercalate into the metal LDH interlayers and adsorb on the surface of electrode under an applied electric field. Since anion adsorption on electrode at applied potentials can be normally expected (see for example J. Energy Chem. 2022, 72, 361), it is necessary to focus on the characterization of whether PF_6^- exhibits intercalation behavior in LDH.
- 2) It's proposed in the manuscript that PF_6^- stabilizes metal sites, an important point to understand the performance improvement. However, authors should provide additional evidence to further substantiate this claim.
- 3) More experimental and computations details are needed. For example, to show the chloride repelling, the authors have employed CPMD simulations to investigate the interactions between anions at the electrolyte-electrode interface. Details of the simulations should be provided. It's stated that the catalyst was activated at a lower positive potential to achieve anionic intercalation while preventing the formation of $\gamma\text{-NiOOH}$. Please provide detailed discussion for this process.
- 4) To highlight the superiority of PF_6^- additive, more comparison of electrocatalytic activity should be given. For example, it's claimed that PF_6^- additive makes the most stable CoFe-based anode. Tabulated performance comparison in terms of the activity as well as the stability metrics would be informative and helpful.
- 5) Authors are suggested to check throughout the main text and SI and perform language polishing.

Reviewer #2

(Remarks to the Author)

This manuscript demonstrates synchronous interlayer and surface engineering using single PF_6^- anion species to achieve Cl^- rejection, with the NiFe LDH anode operating for over 5,000 hours at 1 A cm^{-2} and 2,300 hours at 2 A cm^{-2} . The authors support their work with extensive experimental data and advanced constant-potential molecular dynamics simulations, which confirm the chloride-repelling effect of the PF_6^- anion. Overall, the advancements presented are important for the field of seawater electrolysis, and I recommend publication in Nature Communications. To refine the work further, I have the following suggestions.

- (1). Is immersion alone sufficient to achieve PF_6^- intercalation?
- (2). In Supplementary Fig. 13, while the post-reaction framework is maintained, it should be clarified whether the enhanced ASO stability of the NiFe LDH/NF electrode via electrolyte optimization is also attributable to mechanical stability.
- (3). A comparative analysis should be included to determine if PF_6^- remains in the system after prolonged reaction times.
- (4). The in situ Raman measurements should specify the applied potentials, not just ocp/1.75 V (refer to Fig. 2).
- (5). In the constant-potential molecular dynamics simulations section, the rationale for selecting a carbon substrate instead of

- NiFe LDH should be clarified, along with a detailed discussion of the benefits of constant-potential conditions.
- (6). In the MEA electrolyzer section, a stability comparison using NiFe LDH/NF is suggested.
 - (7). An economic evaluation is recommended.
 - (8). Detailed experimental procedures should be provided, including the source and thickness of the nickel foam, catalyst loading, measurement protocols (with or without catalyst activation), and data analysis methods, such as the IR correction process.
 - (9). Additionally, the authors should further discuss the advantages of their approach over previously reported methods for corrosion-resistant seawater oxidation.

Reviewer #3

(Remarks to the Author)

Comments

- General comment

In this study, the authors explored the application of hexafluorophosphate (PF₆⁻) as an electrolyte additive to mitigate the degradation issues associated with chloride anions during seawater electrolysis. The investigation focused on the behavior of PF₆⁻, which intercalates into the galleries of layered double hydroxides (LDH) and adsorbs onto surfaces during the electrolysis process. The authors propose that PF₆⁻ plays a vital role in effectively obstructing Cl⁻ ions and stabilizing iron, thereby preventing segregation. This stabilization facilitates exceptionally stable long-term performance, achieving 5000 hours at a current density of 1 A cm⁻² and 2300 hours at 2 A cm⁻² using NiFe LDH. Constant potential molecular dynamics (CPMD) simulations provide a theoretical framework illustrating how the accumulation of PF₆⁻ can exclude Cl⁻ ions, thus reducing corrosion. By employing PF₆ as an additive, this study reports significantly enhanced performance compared to recent advancements in the field of seawater electrolysis. Given these substantial improvements, the findings present a notable degree of novelty. However, the manuscript, in its current form, lacks detail in specific areas. With further substantial revisions to address these deficiencies, it may be considered for publication.

1. The authors report a Tafel slope of 12.32 mV dec⁻¹ for the PF₆⁻-containing system; however, this value appears to be exceptionally low, raising concerns regarding its validity. Furthermore, the potential range utilized for calculating the Tafel slope seems rather narrow. It is recommended that the authors provide additional data, such as CA or LSV conducted at a very slow scan rate (e.g., 0.2 mV sec⁻¹), to substantiate their findings. In its current form, it is difficult to eliminate the possibility that the calculated Tafel slope may be influenced by non-OER chemical processes occurring within the specified potential range, which could potentially introduce inaccuracies.
2. In the long-term stability test results shown in Fig. 1d, the potential decreases over time, which is unexpected even after 5000 hours. The authors should provide additional explanations to clarify this phenomenon.
3. The authors have used RuO₂ as a benchmark reference sample for comparison. Although RuO₂ is a highly active catalyst, IrO₂ is more widely recognized as a benchmark catalyst, particularly in terms of both activity and stability. The authors should obtain and include comparative data using IrO₂ to enhance their analysis.
4. In Supplementary Figure 11, it is not clear which values correspond to theoretical calculations and which represent experimental measurements. The authors should review and revise the figure.
5. The authors have not provided any details regarding the experimental conditions for the EIS measurements. A detailed description of the EIS measurement conditions, including parameters such as frequency and applied potential, should be included.
6. In Fig. 2, the authors attributed the peaks observed at 454 cm⁻¹ and 527 cm⁻¹ to the E_g mode of Ni(II)-O and the A_{1g} mode of Ni(II)-O, respectively. However, this peak assignment may be somewhat confusing. Previous studies (e.g., Chem. Sci., 2016, 7, 2639) have assigned these two peaks to the A_{1g} stretching modes of Ni-OH and Ni-O. Additionally, the peak assignments may differ depending on whether the hydroxide phase is α-phase or β-phase. Considering these previous findings, the authors should provide further clarification regarding their peak assignments.
7. The authors provide limited explanation regarding the spectral features observed in the potential-dependent Raman spectra. A more thorough discussion is needed to clarify the observed spectral changes and their implications. In the Raman spectra shown in Figures 2a and 2b, the Ni(III)-O peaks display a blue shift toward higher frequencies as the potential increases in the OER region. This shift indicates a shortening of the Ni-O bond length. Particularly, the peak at 556 cm⁻¹, corresponding to the A_{1g} mode, is associated with oxygen atom vibrations that occur perpendicular to the oxygen plane, which suggests a decrease in interlayer spacing. However, the authors propose that the intercalation of PF₆⁻ increases the interlayer spacing of the LDH, which seems to contradict the Raman spectroscopy results. To resolve this inconsistency, the authors should provide additional experiments or a more detailed explanation to clarify this discrepancy.
8. Previous studies have clearly shown that an increase in the interlayer spacing of LDH results in a red shift of the Raman peaks, as demonstrated by experiments involving cation intercalation. The authors should reference the relevant studies (Angew. Chem. Int. Ed. 2020, 59, 8072–8077; Angew. Chem. Int. Ed. 2019, 58, 12999–13003; Angew. Chem. Int. Ed. 2023, 62, e202313886) and include a further discussion.
9. The assignment of the peak around 300 cm⁻¹ to Fe-O is questionable. This spectral region is more commonly associated with metal-OH vibrations, and it is important to consider that it could correspond to Ni-OH. Previous studies (Energy Technol. 2020, 8, 2000607; ACS Appl. Mater. Interfaces 2020, 12 (38), 42850–42858) have reported similar findings. To clarify the identification of this peak, the authors should provide reference spectra using Ni LDH for comparison.
10. To demonstrate the universality of PF₆⁻, the authors applied it to CoFe LDH and presented performance evaluation results. However, similar to the NiFe LDH experiments, it is important to provide additional operando Raman spectroscopy data and XPS results for CoFe LDH in PF₆⁻-containing electrolytes to strengthen the supporting evidence.
11. Following the long-term stability test, the authors should enhance their analysis beyond merely presenting SEM images.

They should incorporate EDS mapping and XPS measurements of the electrode after testing. This additional data would provide a clearer understanding of the changes in elemental composition in both the bulk and surface regions.

Version 1:

Reviewer comments:

Reviewer #1

(Remarks to the Author)

The authors have well addressed reviewers' comments: (1) The multiple roles of PF6- intercalation into LDH, anion adsorption, and stabilizing metal sites are more clearly demonstrated with solid evidences. (2) More experimental and computational details are provided to make the study more informative. (3) More comparative analysis and performance comparison are given. Thus, from my side, this revised version is publishable.

Reviewer #2

(Remarks to the Author)

The authors have carefully addressed my concerns on this work. The substantial improvements of this work has improved the quality of this work. Therefore, I think this work can be considered for publication now.

Reviewer #3

(Remarks to the Author)

I confirmed the reviewer's are acceptable. So I think the manuscript is now ready for publication following the revisions.

Point-by-Point Responses to Reviewers' Comments

We express our sincere gratitude to the editor and all reviewers for their invaluable feedback, which we have utilized to enhance the quality of our manuscript (NCOMMS-25-03660). The reviewer comments are presented in *italic* and **bold** font, while our responses, including the incorporation of additional figures, tables, descriptions, and other elements, are highlighted in **blue** text.

Point-by-point response to the reviewers #1

Reviewer #1 (Remarks to the Author): In this study, Sun and coworker report the role of hexafluorophosphate as an effective additive to improve direct seawater hydrolysis. They demonstrate that the addition of hexafluorophosphate in electrolytes can intercalate into interlayers of nonprecious transition metal LDH electrocatalysts, adsorb onto electrode surfaces, and accumulate to repel chloride ions. They also combine molecular dynamics simulations and experimental analysis to support the claims. In my opinion, the introduction of anion additive in electrolyte presents a novel, facile approach to well address the stability problem in seawater oxidation, presenting a solid progress in this field that is increasing research interest currently. Thus, I'd like to recommend publication of this work in Nature Communications after minor revision by considering the comments and concerns below.

General Response: We sincerely thank you for your positive assessment of our work and for the constructive comments and suggestions. Your encouraging remarks and thoughtful feedback have been invaluable in guiding the revision process. In response, we have conducted additional characterizations, refined the mechanistic interpretations, and revised the manuscript to improve clarity, coherence, and scientific rigor. We have also thoroughly polished the language throughout the main text and Supplementary Information. We greatly appreciate your insightful feedback, which has helped us further refine and strengthen the manuscript.

Comment 1: *The Authors state that PF_6^- can intercalate into the metal LDH*

interlayers and adsorb on the surface of electrode under an applied electric field. Since anion adsorption on electrode at applied potentials can be normally expected (see for example J. Energy Chem. 2022, 72, 361), it is necessary to focus on the characterization of whether PF_6^- exhibits intercalation behavior in LDH.

Response 1: We sincerely appreciate your valuable comment. As you pointed out, surface adsorption of anions on electrode at applied potentials is normally expected (J. Energy Chem. 2022, 72, 361). To clarify the potential-dependent behavior of PF_6^- , we further conducted operando Raman spectroscopy on CoFe LDH/NF in PF_6^- -containing seawater. The spectra reveal evidence that PF_6^- follows a potential-driven intercalation-to-adsorption for CoFe LDH (**Fig. R1**; Supplementary Fig. 31). These results are consistent with our observations on NiFe LDH and reinforce the dual-mode interaction (intercalation and adsorption) of PF_6^- for LDH. Detailed analysis of the operando Raman results has been provided in the revised Supplementary Information.

Fig. R1 Operando Raman spectra of CoFe LDH in PF_6^- -containing seawater during ASO.

Comment 2: *It's proposed in the manuscript that PF_6^- stabilizes metal sites, an important point to understand the performance improvement. However, authors should provide additional evidence to further substantiate this claim.*

Response 2: We sincerely appreciate your insightful comment regarding the

stabilizing role of PF_6^- on metal sites. In the original manuscript, we provided quantitative evidence of Fe stabilization via ICP-MS and TOF-SIMS. After 120 hours of chronoamperometry tests at 1 A cm^{-2} , the Fe leaching in PF_6^- -free seawater reached

Fig. R2 (a) HAADF-STEM image and (b–f) corresponding elemental mapping images of NiFe LDH after stability test in PF_6^- -containing seawater.

Fig. R3 XPS spectra of NiFe LDH/NF in the (a) P 2p and (b) F 1s regions after stability test in PF_6^- -containing seawater. XPS spectra of NiFe LDH/NF in the (c) Ni 2p and (d)

Fe 2p regions before and after stability test in PF₆⁻-containing seawater.

1.5 μg mL⁻¹, much higher than 0.291 μg mL⁻¹ in the PF₆⁻-containing seawater (Supplementary Fig. 25). TOF-SIMS mapping (Fig. 2g) further confirms Fe retention, strong PF₆⁻ signals, and minimal Cl⁻, supporting the Cl⁻-repelling and Fe-stabilizing effect of PF₆⁻. To substantiate this claim, we further performed EDS elemental mapping and XPS analysis. EDS mapping (Fig. R2; Supplementary Fig. 28) and XPS analyses (Fig. R3; Supplementary Fig. 29) conducted after the long-term test confirm the presence of P, F, Ni, Fe, and O throughout the nanosheets, alongside extensive NiOOH formation at the surface, also supporting that PF₆⁻ incorporation stabilizes Fe and suppresses its leaching. These results collectively support the critical role of PF₆⁻ in stabilizing metal sites and mitigating Fe dissolution.

Comment 3: More experimental and computations details are needed. For example, to show the chloride repelling, the authors have employed CPMD simulations to investigate the interactions between anions at the electrolyte-electrode interface. Details of the simulations should be provided. It's stated that the catalyst was activated at a lower positive potential to achieve anionic intercalation while preventing the formation of γ-NiOOH. Please provide detailed discussion for this process.

Response 3: We sincerely appreciate your valuable suggestions. As recommended, we have now included detailed descriptions of the computational methods in the revised *Methods* section. “The simulation procedure began by arranging 40 explicit water molecules to create a density of 1 g cm⁻³. Following this, three Cl⁻ were placed randomly in the aqueous phase to ensure uniform distribution. Finally, we achieved the variation of different concentrations by introducing different numbers of PF₆⁻ on the surface of the catalyst.” These simulation parameters are now described in the main text. Additionally, we now provide the experimental details used in the operando Raman tests from open-circuit potential (OCP) to 1.75 V, in 0.05 V increments. Details regarding the activation step are also added in the *Methods*

section: “NiFe LDH/NF electrode was held at 1.15 V for 10 min in PF₆⁻-containing seawater, with Hg/HgO and graphite serving as reference and counter electrodes, respectively. The sample was then removed, rinsed with deionized water, and dried”. This relatively low positive potential can ensure PF₆⁻ intercalation while avoiding the formation of γ-NiOOH during the activation process. These additions are now also reflected in the revised *Methods* section.

Comment 4: To highlight the superiority of PF₆⁻ additive, more comparison of electrocatalytic activity should be given. For example, it's claimed that PF₆⁻ additive makes the most stable CoFe-based anode. Tabulated performance comparison in terms of the activity as well as the stability metrics would be informative and helpful.

Response 4: We appreciate your insightful suggestion. As suggested, we compared the activity and durability of CoFe LDH/NF in PF₆⁻-containing and PF₆⁻-free seawater. Polarization curve reveals that CoFe LDH/NF in PF₆⁻-containing seawater reaches a current density of 1 A cm⁻² at an overpotential of 373 mV (**Fig. R4**; Supplementary Fig. 33), placing it among the most efficient CoFe-based anodes reported to date (**Table R1**; Supplementary Table 5). For stability evaluation, chronopotentiometry tests at 2.0 A cm⁻² shows that in PF₆⁻-containing seawater, the electrode remained stable for over 1,200 hours, whereas the PF₆⁻-free counterpart failed after 65 hours (Fig. 3a), corresponding to an 18.5-fold improvement in durability. This operational lifetime under high current density also renders CoFe LDH/NF as the most durable CoFe-based anode for ASO reported to date (**Table R2**, Supplementary Table 6). These comparison data have been added to the revised manuscript and Supplementary Information.

Fig. R4. (a) Polarization curves and (b) comparison of overpotential of CoFe LDH/NF measured in PF₆⁻-free versus PF₆⁻-containing seawater.

Table R1. Comparison of the overpotentials of CoFe LDH/NF anode in PF₆⁻-containing seawater with recently reported CoFe-based anodes.

Anodes	Electrolyte	j (mA cm ⁻²)	Overpotential (mV)	Reference
CoFe LDH/NF (with KPF ₆ as electrolyte additive)	1 M KOH + seawater + 20 mM PF ₆ ⁻	200	230	This work
		500	307	
		1000	373	
CF@CF-phy/NF	1 M KOH + seawater	500	330	ACS Nano 19 , 1530–1546 (2025)
CoFeAl LDH/NF	20wt.% NaOH + satu. NaCl	10	256	Nat. Commun. 15 , 4712 (2024)
		200	~320	
CoFe-Ci@GQD	1 M KOH + 0.5 M NaCl	100	255	Nat. Sustain. 7 , 158–167 (2024)
Ir/CoFe LDH	6 M NaOH + 2.8 M NaCl	10	202	Nat. Commun. 15 1973 (2024)
Cr-CoFe LDH/NF	1 M KOH + seawater	500	334	Small 20 , 2307294 (2024)
B-Co ₂ Fe LDH/NF	1 M KOH + seawater	100	310	Nano Energy 83 , 105838 (2021)
		500	376	
CoCO ₃ /CoFe LDH/NF	1 M KOH + seawater	500	316	Small 21 , 2409627 (2025)
RuCo-CoFe ₂ O ₄ @IF	1 M KOH + seawater	1000	425	Chem. Eng. J. 503 , 158346 (2025)

Table R2. Comparison of the stability of CoFe LDH/NF anode in PF₆⁻-containing seawater with recently reported CoFe-based anodes.

Anodes	Electrolyte	j (mA cm ⁻²)	Stability (h)	Reference
CoFe LDH/NF (with KPF ₆ as electrolyte additive)	1 M KOH + seawater + 20 mM PF ₆ ⁻	2000	1200	This work
CF@CF-phy/NF	1 M KOH + seawater	1000	1000	ACS Nano 19 , 1530–1546 (2025)
CoFeAl LDH/NF	20wt.% NaOH + satu. NaCl	1000	500	Nat. Commun. 15 , 4712 (2024)
		2000	350	

CoFe-Ci@GQD	1 M KOH + 0.5 M NaCl	1250	2800	Nat. Sustain. 7 , 158–167 (2024)
Ir/CoFe LDH	6 M NaOH + 2.8 M NaCl	800	1000	Nat. Commun. 15 1973 (2024)
Cr-CoFe LDH/NF	1 M KOH + seawater	500	100	Small 20 , 2307294 (2024)
B-Co ₂ Fe LDH/NF	1 M KOH + seawater	500	100	Nano Energy 83 , 105838 (2021)
CeO _{2-x} @CoFe LDH/NF	1 M KOH + 0.5 M NaCl	50	35	Inorg. Chem. Front. 7 , 4461–4468 (2020)
CoCO ₃ /CoFe LDH/NF	1 M KOH + seawater	1000	1000	Small 21 , 2409627 (2025)
CoFePBA/Co ₂ P	20wt.% NaOH + satu. NaCl	1000	1000	Angew. Chem. Int. Ed. 62 , e202309882 (2023)
		2000	100	
CoFe-Ni ₂ P/NF	1 M KOH + seawater	500	500	Adv. Energy Mater. 13 , 2301475 (2023)
FCDs/FeCoSe-VSe/NF	1 M KOH + seawater	200	200	Appl. Surf. Sci. 680 , 161456 (2025)
RuCo-CoFe ₂ O ₄ @IF	1 M KOH + seawater	1000	150	Chem. Eng. J. 503 , 158346 (2025)

Comment 5: Authors are suggested to check throughout the main text and SI and perform language polishing.

Response 5: Thank you for your suggestion. We have carefully reviewed the entire main text and Supplementary Information and have thoroughly polished the language throughout the manuscript

Point-by-point response to the reviewers #2

Reviewer #2 (Remarks to the Author): *This manuscript demonstrates synchronous interlayer and surface engineering using single PF_6^- anion species to achieve Cl^- rejection, with the NiFe LDH anode operating for over 5,000 hours at 1 A cm^{-2} and 2,300 hours at 2 A cm^{-2} . The authors support their work with extensive experimental data and advanced constant-potential molecular dynamics simulations, which confirm the chloride-repelling effect of the PF_6^- anion. Overall, the advancements presented are important for the field of seawater electrolysis, and I recommend publication in Nature Communications. To refine the work further, I have the following suggestions.*

General Response: We sincerely thank you for your positive assessment of our work and for your constructive suggestions. We have carefully addressed each of your comments in our revised manuscript. Where appropriate, we have added new data and clarified methodological details to improve the clarity and completeness of the work. We greatly appreciate your insightful feedback, which has helped us further refine and strengthen the manuscript.

Comment 1: *Is immersion alone sufficient to achieve PF_6^- intercalation?*

Fig. R5. XRD patterns of NiFe LDH/NF after immersion in PF_6^- -containing solution for 0, 0.5 h, and 3 h.

Response 1: Thank you for this valuable comment. To verify whether immersion alone is sufficient to induce PF_6^- intercalation, we performed XRD measurements on NiFe LDH/NF after immersion in PF_6^- -containing solution for 0, 0.5, and 3 hours (**Fig.**

R5). A gradual increase in interlayer spacing was also observed with increasing immersion time, indicating PF_6^- intercalation. Compared to voltage-driven intercalation (10 min), the soaking process proceeds at a slower rate.

Comment 2: In Supplementary Fig. 13, while the post-reaction framework is maintained, it should be clarified whether the enhanced ASO stability of the NiFe LDH/NF electrode via electrolyte optimization is also attributable to mechanical stability.

Response 2: Thank you for your suggestion. While PF_6^- primarily functions as a chloride-repelling agent, our results suggest that mechanical stability also plays a role in the enhanced ASO durability. SEM and structural analyses (Supplementary Figs. 26 and 27) show that the NiFe LDH/NF electrode maintains its nanosheet morphology and substrate integrity after 120 hours in PF_6^- -containing seawater, whereas substantial structural collapse occurs without PF_6^- . This indicates that electrolyte-induced structural preservation contributes to the long-term electrochemical performance. We have clarified this point in the revised version.

Comment 3: A comparative analysis should be included to determine if PF_6^- remains in the system after prolonged reaction times.

Fig. R6 Raman spectrum of NiFe LDH/NF after stability test in PF_6^- -containing seawater.

Response 3: Thank you for raising this point. To determine if PF_6^- remains in the system after prolonged reaction times, we have conducted EDS elemental mapping, XPS analysis, and Raman spectroscopy after long-term electrolysis. The results

confirm that PF_6^- is still detectable after prolonged electrolysis (Fig. R2, R3, R6; Supplementary Fig. 28–30). These data have been incorporated into the revised version.

Comment 4: The in situ Raman measurements should specify the applied potentials, not just ocp/1.75 V (refer to Fig. 2).

Response 4: Thank you for raising this point. In accordance with your suggestion, we have revised the manuscript to specify that the in situ Raman measurements were conducted from open-circuit potential to 1.75 V, with an incremental step of 0.05 V. This information has been clarified in the *Methods* section. We have also added annotations in Fig. 2, and Supplementary Fig. 31.

Comment 5: In the constant-potential molecular dynamics simulations section, the rationale for selecting a carbon substrate instead of NiFe LDH should be clarified, along with a detailed discussion of the benefits of constant-potential conditions.

Response 5: We appreciate your insightful comment. Given that this is a versatile method and our primary focus is on the repulsive interactions between PF_6^- and Cl^- at the interface, we selected carbon as the catalyst. Additionally, this approach can substantially reduce computational costs. Furthermore, constant potential molecular dynamics (CPMD) is an effective methodology for examining the dynamics of electrochemical interfaces in explicit solvent environments under fixed potential conditions. This approach is well-suited for systems of moderate size, allowing comprehensive investigation of interfacial properties and electrochemical interactions.

Comment 6: In the MEA electrolyzer section, a stability comparison using NiFe LDH/NF is suggested.

Response 6: Thank you for this suggestion. A direct stability comparison has been conducted using the NiFe LDH/NF||Pt/C/NF electrolyzer. In PF_6^- -containing seawater, the system operated stably for over 1,000 hours at 1.0 A cm^{-2} (Fig. 4c), whereas in PF_6^- -free seawater, rapid degradation occurred after only 31 hours (Fig. R7;

Supplementary Fig. 34). These results further confirm the critical role of PF_6^- in sustaining electrode stability under prolonged operating conditions.

Fig. R7 Chronopotentiometry curve of the NiFe LDH/NF||Pt/C/NF electrolyzer conducted in PF_6^- -free seawater at a j of 1.0 A cm^{-2} .

Comment 7: An economic evaluation is recommended.

Response 7: Thank you for this suggestion. An economic evaluation has been added to the revised manuscript, showing that at an electricity price of $\text{US}\$0.02 \text{ kWh}^{-1}$, the hydrogen production cost is approximately $\text{US}\$1.07$ per kg H_2 at 1.0 A cm^{-2} and 2.04 V . Details of the calculation, are provided in Supplementary Note 3. “The NiFe LDH/NF||Pt/C/NF flow-type electrolyzer operated under a dual-feed configuration using $6 \text{ M KOH} + \text{seawater}$ as both the anolyte and catholyte, with PF_6^- introduced into the anolyte. The energy consumption was evaluated under electrolysis conditions of 1.0 A cm^{-2} , 2.02 V , $60 \text{ }^\circ\text{C}$, and an electrode area of $1 \times 1 \text{ cm}^2$. Under these conditions, the power input was calculated to be 2.02 W cm^{-2} . The hydrogen production rate was determined to be $5.18 \times 10^{-6} \text{ mol cm}^{-2} \text{ s}^{-1}$. Taking the lower heating value (LHV) of H_2 as 120 MJ kg^{-1} (equivalent to $241.9 \text{ kJ mol}^{-1}$), the output power density associated with H_2 generation was estimated to be 1.253 W cm^{-2} . Accordingly, the energy efficiency (based on LHV) was calculated to be 62.0% . The electricity consumption per kilogram of hydrogen was further estimated as $53.7 \text{ kWh kg}^{-1} \text{ H}_2$ (4.49 kWh m^{-3}), corresponding to a production cost of $\text{US}\$1.07 \text{ kg}^{-1} \text{ H}_2$, when utilizing offshore renewable electricity at $\text{US}\$0.02 \text{ kWh}^{-1}$ ”.

Comment 8: Detailed experimental procedures should be provided, including the source and thickness of the nickel foam, catalyst loading, measurement protocols (with or without catalyst activation), and data analysis methods, such as the IR correction process.

Response 8: Thank you for raising these important points. We have added the experimental details in the revised *Methods* section. Specifically, nickel foam (NF, 0.2 mm thickness) was purchased from Qingyuan Metal Materials Co., Ltd. (Xingtai, China). The catalyst loading of NiFe LDH was approximately 1.93 mg cm⁻². We have also included detailed descriptions of the catalyst activation procedure, measurement protocols. In addition, figure captions for all relevant data now clearly indicate whether iR correction was applied.

Comment 9: Additionally, the authors should further discuss the advantages of their approach over previously reported methods for corrosion-resistant seawater oxidation.

Response 9: Thank you for your valuable suggestion. As suggested, we have included a comparative discussion in the revised manuscript to emphasize how our strategy differs from and improves upon previously reported approaches. Prior methods, such as the use of surface-adsorbed oxyanions (SO₄²⁻, PO₄³⁻), formation of insoluble chloride-repellent coatings (e.g., AgCl), or dual-component surface/interlayer modification that have made progress in enhancing chloride resistance. However, these strategies often involve multi-step fabrication processes or limited durability under extended electrolysis. Our approach introduces a single, non-oxygen PF₆⁻ anion species that enables both interlayer intercalation and surface adsorption under an applied field, effectively repelling Cl⁻ and achieving over 5,000 hours of stable operation. The simplicity and efficacy of this method address a key limitation. These distinctions have been discussed in the revised *Introduction* and *Results* sections.

Point-by-point response to the reviewers #3

Reviewer #3 (Remarks to the Author): *In this study, the authors explored the application of hexafluorophosphate (PF_6^-) as an electrolyte additive to mitigate the degradation issues associated with chloride anions during seawater electrolysis. The investigation focused on the behavior of PF_6^- , which intercalates into the galleries of layered double hydroxides (LDH) and adsorbs onto surfaces during the electrolysis process. The authors propose that PF_6^- plays a vital role in effectively obstructing Cl^- ions and stabilizing iron, thereby preventing segregation. This stabilization facilitates exceptionally stable long-term performance, achieving 5000 hours at a current density of 1 A cm^{-2} and 2300 hours at 2 A cm^{-2} using NiFe LDH. Constant potential molecular dynamics (CPMD) simulations provide a theoretical framework illustrating how the accumulation of PF_6^- can exclude Cl^- ions, thus reducing corrosion. By employing PF_6^- as an additive, this study reports significantly enhanced performance compared to recent advancements in the field of seawater electrolysis. Given these substantial improvements, the findings present a notable degree of novelty. However, the manuscript, in its current form, lacks detail in specific areas. With further substantial revisions to address these deficiencies, it may be considered for publication.*

General Response: We deeply appreciate your encouraging remarks and constructive suggestions. Your insights have been instrumental in refining the manuscript and ensuring that our findings are presented more clearly and rigorously. In response to your suggestions, we have conducted careful revisions and clarified several aspects of our analysis. Thank you again for your thoughtful review, which has been very helpful in improving the overall quality and completeness of our work.

Comment 1: *The authors report a Tafel slope of $12.32 \text{ mV dec}^{-1}$ for the PF_6^- -containing system; however, this value appears to be exceptionally low, raising concerns regarding its validity. Furthermore, the potential range utilized for calculating the Tafel slope seems rather narrow. It is recommended that the authors provide additional data, such as CA or LSV conducted at a very slow scan*

rate (e.g., 0.2 mV sec^{-1}), to substantiate their findings. In its current form, it is difficult to eliminate the possibility that the calculated Tafel slope may be influenced by non-OER chemical processes occurring within the specified potential range, which could potentially introduce inaccuracies.

Fig. R8 Polarization curves measured at a slow scan rate of 0.2 mV s^{-1} for NiFe LDH/NF in PF_6^- -containing seawater with (a) forward scan and (b) reverse scan.

Fig. R9 (a) Polarization curves and (b) corresponding Tafel slopes of NiFe LDH/NF in PF_6^- -free and PF_6^- -containing seawater, alongside RuO_2/NF , IrO_2/NF and NF in PF_6^- -free seawater at a slow scan rate of 1 mV s^{-1} .

Response 1: Thank you for your detailed comments. To address the concern regarding the validity of the reported Tafel slope and potential interference from non-OER processes, we performed LSV tests at a very slow scan rate of 0.2 mV s^{-1}

using a CHI 660E workstation. The current response under these conditions, however, was unstable, showing no significant increase in the forward scan or decrease in the reverse scan (**Fig. R8**), and thus was not suitable for accurate Tafel slope analysis. As an alternative, we employed a scan rate of 1 mV s^{-1} , which provided reliable polarization curves (**Fig. R9**; Supplementary Fig. 10). These data support that the Tafel slope is not significantly affected by non-OER chemical processes. Furthermore, we recalculated the Tafel slope using a broader potential range, and the results demonstrate enhanced OER kinetics for NiFe LDH/NF in PF_6^- -containing seawater, outperforming benchmark RuO_2 and IrO_2 catalysts. These updated results have been incorporated into the revised manuscript.

Comment 2: *In the long-term stability test results shown in Fig. 1d, the potential decreases over time, which is unexpected even after 5000 hours. The authors should provide additional explanations to clarify this phenomenon.*

Response 2: Thank you for your comment. The observed decrease in potential over time during the long-term stability test (Fig. 1d) is primarily attributed to an increase in electrolyte concentration as a result of water consumption during electrolysis. As water is gradually consumed, the effective alkalinity of the electrolyte increases, which can reduce the overpotential required to sustain the same current density, even though the alkaline seawater was replenished periodically rather than continuously in three-electrode system. Additionally, seasonal temperature variations may have contributed to this effect. The test was conducted in Chengdu, China, from March through the summer months, experienced higher ambient temperatures from July onward, which likely enhanced conductivity and reaction kinetics, further contributing to the observed potential decline.

Comment 3: *The authors have used RuO_2 as a benchmark reference sample for comparison. Although RuO_2 is a highly active catalyst, IrO_2 is more widely recognized as a benchmark catalyst, particularly in terms of both activity and stability. The authors should obtain and include comparative data using IrO_2 to*

enhance their analysis.

Response 3: Thank you for your helpful suggestion. As IrO₂ is widely recognized as a benchmark catalyst for alkaline OER, we have added comparative data using commercial IrO₂ in the revised manuscript. As shown in Fig. 1a, Fig. R9, Supplementary Fig. 10, and Fig. 1b, the NiFe LDH/NF electrode in PF₆⁻-containing seawater exhibits significantly lower overpotentials at current densities of 200, 500, and 1000 mA cm⁻² (209, 247, and 282 mV, respectively) than the PF₆⁻-free system (282, 315, and 343 mV), and also outperforms both RuO₂ and IrO₂ reference catalysts. In addition, the Tafel slope analysis confirms accelerated OER kinetics in the presence of PF₆⁻. These new results have been incorporated into the revised manuscript.

Comment 4: In Supplementary Figure 11, it is not clear which values correspond to theoretical calculations and which represent experimental measurements. The authors should review and revise the figure.

Response 4: Thank you for bringing this to our attention. To improve clarity, we have revised the figure and updated it as Fig. R10 (Supplementary Fig. 12) in the revised manuscript. Theoretical values are now clearly labeled in red and experimental values in blue, ensuring that the distinction between the two data sets is unambiguous.

Fig. R10 Comparison of collected O₂ with theoretical values for NiFe LDH/NF in PF₆⁻-containing seawater at 1 A cm⁻².

Comment 5: The authors have not provided any details regarding the experimental conditions for the EIS measurements. A detailed description of the EIS measurement conditions, including parameters such as frequency and applied potential, should be included.

Response 5: We have now included a detailed description of the EIS measurement conditions in the revised Supplementary Information (Supplementary Fig. 11). Specifically, Nyquist plots were recorded at applied potentials of 1.405, 1.415, 1.435, 1.455, 1.475, and 1.485 V (vs. RHE), across a frequency range from 10^{-2} to 10^5 Hz.

Comment 6: In Fig. 2, the authors attributed the peaks observed at 454 cm^{-1} and 527 cm^{-1} to the E_g mode of Ni(II)-O and the A_{1g} mode of Ni(II)-O, respectively. However, this peak assignment may be somewhat confusing. Previous studies (e.g., Chem. Sci., 2016, 7, 2639) have assigned these two peaks to the A_{1g} stretching modes of Ni-OH and Ni-O. Additionally, the peak assignments may differ depending on whether the hydroxide phase is α -phase or β -phase. Considering these previous findings, the authors should provide further clarification regarding their peak assignments.

Response 6: We sincerely appreciate your insightful comment regarding the Raman peak assignments. In light of your suggestion and previous reports (e.g., Chem. Sci. 2016, 7, 2639), we have re-assigned the peaks at 454 cm^{-1} and 527 cm^{-1} to the stretching modes of Ni(II)-OH and Ni(II)-O, respectively. This assignment is also supported by recent studies (Chem Catal. 2023, 3, 100475; Angew. Chem. Int. Ed. 2020, 59, 8072–8077). These updates have been incorporated into Fig. 2a–c and Supplementary Figs. 20 and 30. We thank you again for helping us improve the accuracy and clarity of our spectral interpretation.

Comment 7: The authors provide limited explanation regarding the spectral features observed in the potential-dependent Raman spectra. A more thorough discussion is needed to clarify the observed spectral changes and their implications. In the Raman spectra shown in Figures 2a and 2b, the Ni(III)-O peaks display a blue

shift toward higher frequencies as the potential increases in the OER region. This shift indicates a shortening of the Ni-O bond length. Particularly, the peak at 556 cm⁻¹, corresponding to the A_{1g} mode, is associated with oxygen atom vibrations that occur perpendicular to the oxygen plane, which suggests a decrease in interlayer spacing. However, the authors propose that the intercalation of PF₆⁻ increases the interlayer spacing of the LDH, which seems to contradict the Raman spectroscopy results. To resolve this inconsistency, the authors should provide additional experiments or a more detailed explanation to clarify this discrepancy.

Response 7: We appreciate your thoughtful comment regarding the interpretation of the potential-dependent Raman spectra. We agree that the blue shift of the Ni(III)-O peaks, particularly the 556 cm⁻¹ A_{1g} mode, reflects a shortening of the Ni-O bond and a corresponding decrease in interlayer spacing. We apologize for not conveying this distinction clearly in the original manuscript. To clarify, the discussion of increased interlayer spacing pertains specifically to PF₆⁻ intercalation observed after the NiFe LDH/NF electrode was held at 1.15 V for 10 minutes in PF₆⁻-containing seawater. Operando Raman spectra (Fig. 2b; Supplementary Fig. 14) show the emergence of the E_g band and disappearance of CO₃²⁻/NO₃⁻ signals, indicating that PF₆⁻ intercalation occurs under low potentials. Supporting XRD, TEM, and Raman results (Supplementary Figs. 15–19) collectively confirm successful interlayer incorporation of PF₆⁻. Subsequently, operando Raman spectra of PF₆⁻-intercalated NiFe LDH in PF₆⁻-free seawater (Fig. 2c) reveal a progressive blue shift of the Ni(III)-O band with increasing potential, which is consistent with partial PF₆⁻ de-intercalation and interlayer contraction. We also apologize for the copy-paste error in Fig. 2b, where the fourth and fifth spectra were inadvertently duplicated. This has now been corrected. These clarifications have been incorporated into the revised manuscript, and we thank you for helping us improve the accuracy and consistency of our interpretation.

Comment 8: *Previous studies have clearly shown that an increase in the interlayer spacing of LDH results in a red shift of the Raman peaks, as demonstrated by*

experiments involving cation intercalation. The authors should reference the relevant studies (Angew. Chem. Int. Ed. 2020, 59, 8072–8077; Angew. Chem. Int. Ed. 2019, 58, 12999–13003; Angew. Chem. Int. Ed. 2023, 62, e202313886) and include a further discussion.

Fig. R11 Comparison of Raman spectra of PF₆⁻-intercalated NiFe LDH and NiFe LDH at OCP.

Fig. R12 First derivative Ni K-edge spectra of PF₆⁻-intercalated NiFe LDH and NiFe LDH, NiO, and Ni foil.

Response 8: Thank you for your valuable comment. Prior studies have demonstrated that the intercalation of large alkali cations into layered double hydroxides (LDHs) typically leads to a red shift in the Raman-active Ni–O vibrational modes, primarily attributed to elongation of Ni–O bonds and increased interlayer spacing (Angew. Chem. Int. Ed. 2020, 59, 8072–8077; Angew. Chem. Int. Ed. 2019, 58, 12999–13003; Angew. Chem. Int. Ed. 2023, 62, e202313886). To further examine this relationship,

we compared the Raman spectra of NiFe LDH before and after PF_6^- intercalation (Fig. R11; Supplementary Fig. 21). In contrast to cation intercalation, PF_6^- intercalation induces a slight blue shift ($\sim 3.56 \text{ cm}^{-1}$), rather than a red shift. This observation is corroborated by first-derivative Ni K-edge X-ray absorption spectra (Fig. R12; Supplementary Fig. 24), which reveal a subtle contraction of the Ni–O bond length following PF_6^- incorporation.

Comment 9: *The assignment of the peak around 300 cm^{-1} to Fe–O is questionable. This spectral region is more commonly associated with metal–OH vibrations, and it is important to consider that it could correspond to Ni–OH. Previous studies (Energy Technol. 2020, 8, 2000607; ACS Appl. Mater. Interfaces 2020, 12 (38), 42850–42858) have reported similar findings. To clarify the identification of this peak, the authors should provide reference spectra using Ni LDH for comparison.*

Response 9: We sincerely thank you for highlighting this important point. After reviewing the relevant literature (Energy Technol. 2020, 8, 2000607; ACS Appl. Mater. Interfaces 2020, 12, 42850–42858) and considering your suggestion, we conducted Raman measurement on Ni LDH (Fig. R13). A comparable peak near 300 cm^{-1} was also observed, supporting its assignment to Ni–OH lattice vibrations rather than Fe–O. Accordingly, we have revised the peak assignment in the manuscript. We appreciate your comment, which has helped us improve the rigor and accuracy of our spectral interpretation.

Fig. R13 Raman spectrum of Ni LDH.

Comment 10: *To demonstrate the universality of PF_6^- , the authors applied it to*

CoFe LDH and presented performance evaluation results. However, similar to the NiFe LDH experiments, it is important to provide additional operando Raman spectroscopy data and XPS results for CoFe LDH in PF₆⁻-containing electrolytes to strengthen the supporting evidence.

Response 10: Thank you for your thoughtful feedback. To validate the intercalation and surface adsorption behavior of PF₆⁻ in CoFe LDH, we conducted operando Raman measurements in PF₆⁻-containing seawater from open-circuit potential (OCP) to 1.70 V vs. RHE with a step of 0.05 V (**Fig. R1**; Supplementary Fig. 31). “Operando Raman spectra of CoFe LDH in PF₆⁻-containing seawater were collected from OCP to 1.70 V vs. RHE (step: 0.05 V) to investigate PF₆⁻ intercalation and surface adsorption behaviors. The spectrum displays signals at 482, 523, and 698 cm⁻¹, corresponding to the E_g, F_{2g}, and A_{1g} vibrations of the symmetric stretching mode of Co–O and the bending mode of O–Co–O (*CrystEngComm* **24**, 6018–6030 (2022); *Chem. Mater.* **32**, 4303 (2020); *ACS Catal.* **8**, 1238–1247 (2018); *Chem. Eng. J.* **472**, 145076 (2023)). As the potential increases, the A_{1g} and F_{2g} bands and the Co–OH lattice signal at 300 cm⁻¹ gradually vanish, and the shoulder for E_g and F_{2g} modes at 464 and 536 cm⁻¹ appears (partially masked by PF₆⁻ signals), indicating the formation of CoOOH. Interestingly, the 482 cm⁻¹ E_g mode (overlapping with PF₆⁻ vibrational modes) initially increases in intensity (which was expected to decrease monotonically), along with the fading of the 727 cm⁻¹ CO₃²⁻/NO₃⁻ peak, suggesting PF₆⁻ intercalation. At higher potentials, the 482 cm⁻¹ mode weakens, indicating partial anion deintercalation, and the 727 cm⁻¹ A_{1g} vibration of PF₆⁻ emerges and intensifies, implying surface adsorption and accumulation of PF₆⁻ under an applied electric field. These results support that PF₆⁻ follows a potential-driven intercalation and adsorption for CoFe LDH, consistent with the observations for NiFe LDH”. In addition, XPS spectra of the electrode after testing (**Fig. R14**; Supplementary Fig. 32) clearly show signals in both the P 2p and F 1s regions, further confirming the incorporation of PF₆⁻. These findings have been included in the revised manuscript as supporting evidence. We sincerely thank you for your valuable suggestion.

Comment 11: Following the long-term stability test, the authors should enhance their analysis beyond merely presenting SEM images. They should incorporate EDS mapping and XPS measurements of the electrode after testing. This additional data would provide a clearer understanding of the changes in elemental composition in both the bulk and surface regions.

Response 11: Thank you for your valuable suggestion. To enhance our post-stability analysis, we have performed EDS mapping and XPS characterization of the electrode after long-term operation to analyze elemental composition in both the bulk and surface regions. EDS mapping and XPS analyses (Fig. R2, R3; Supplementary Fig. 28, 29) conducted after the long-term test confirm the presence of P, F, Ni, Fe, and O throughout the nanosheets, alongside extensive NiOOH formation at the surface, also supporting that PF_6^- incorporation stabilizes Fe and suppresses its leaching. These data have been incorporated into the revised version.

Fig. R14 XPS spectra of NiFe LDH/NF in the (a) P 2p and (b) F 1s regions after 120 hours of stability test in PF_6^- -containing seawater. XPS spectra of NiFe LDH/NF in the (c) Ni 2p and (d) Fe 2p regions before and after stability test in PF_6^- -containing seawater.